# Social Hierarchy-Guided Evolutionary Neural Architecture Search for Efficient and Automated Design

## Abstract

Neural Architecture Search (NAS) serves as an important component in Automated Machine Learning. Compared with reinforcement learning and gradient-based NAS approaches, evolutionary computation-based NAS (ENAS) has gained prominence due to its lower dependence on domain expertise and superior adaptability across diverse problem domains. However, despite a lot of research, how to significantly reduce the computational cost while pursuing high accuracy is still a huge challenge for ENAS. To address this issue, we propose a Social Hierarchy-guided Evolutionary Neural Architecture Search algorithm (SH-ENAS). In this algorithm, inspired by the social hierarchy, a novel population organization structure is designed, and based on it, effective guidance operations are designed for the subsequent evolutionary search process. Additionally, to further reduce computational overhead, a progressive evaluation search method is proposed, which introduces weight inheritance and validation-loss-guided early stopping operation to prevent unnecessary evaluations of the architecture. The experimental results demonstrate that SH-ENAS achieves test errors of $2.50\%$ and $16.24\%$ on CIFAR-10 and CIFAR-100, respectively, outperforming existing state-of-the-art methods. In particular, SH-ENAS requires only 10 population individuals and 12 iterations to complete the search, with computational costs as low as 0.69 GPU days and 0.83 GPU days, validating the significant advantages of the new algorithm in terms of accuracy, computational efficiency, and automation.

## 1 Introduction

Neural Architecture Search (NAS) represents a critical component of Automated Machine Learning (AutoML), focusing on the automated design of neural networks to reduce the cost and complexity associated with manual architecture development. It has demonstrated significant success across various domains, including image classification [1, 2, 3, 4, 5], object detection[6, 7, 8, 9, 10], and semantic segmentation[11, 12, 13, 14, 15, 16].

Existing NAS algorithms can be categorized into three dominant paradigms based on optimization mechanisms: Reinforcement Learning-based NAS (RL-NAS), Gradient-based NAS (GD-NAS), and Evolutionary computation-based NAS (ENAS)[17].While RL-NAS has achieved remarkable performance through iterative policy optimization, its computational cost remains prohibitively high. Even for medium-scale benchmarks such as CIFAR-10, it typically requires more than 2000 GPU days. Gradient-based approaches, represented by Differentiable architecture search (DARTS) [18], improve efficiency by progressively relaxing the search space. However, their effectiveness heavily relies on carefully designed supernet architectures, which demand substantial domain expertise for topological initialization and hyperparameter tuning.

Submitted to 39th Conference on Neural Information Processing Systems (NeurIPS 2025). Do not distribute.

In comparison, ENAS demonstrates distinct advantages in adaptability through its population-based evolutionary mechanism. By implementing dynamic random crossover and mutation operations, ENAS reduces prior knowledge dependency while maintaining competitive performance. These characteristics make it particularly suitable for multi-modal tasks and complex task scenarios.

Nevertheless, ENAS still faces critical challenges in terms of computational scalability. Even with acceleration techniques such as surrogate models and weight inheritance, it requires substantial computational resources, which significantly limits its deployment in resource-constrained environments. This computational overhead stems from the inherent exploration–exploitation trade-off in evolutionary algorithms and remains an open research problem hindering their widespread adoption in practical applications.

In general, the typical ENAS framework comprises four key steps, including population initialization, encoding, performance evaluation, and an evolutionary search process involving crossover and mutation operations, which are commonly implemented using Genetic Algorithms (GA) [19].To mitigate the high computational cost of ENAS, extensive work has been conducted, as shown in Section 2, primarily focusing on reducing the overhead associated with NAS performance evaluation. Nevertheless, how to reduce the computational resource consumption of ENAS while maintaining search accuracy still remains a significant challenge.

It should be pointed out that, in addition to the performance evaluation overhead, another critical factor that leads to the high computational cost of ENAS, which is often underestimated in most current studies, lies in its evolutionary search process, which can significantly increase resource consumption due to insufficient search capability and inefficiency. This is mainly due to two factors:

1) **Lack of effective guidance in evolution**: GA used in current ENAS employs fixed-probability random operations for crossover and mutation. This leads to a lack of effective guidance in evolution, severely limiting the algorithm's global search ability. As a result, it requires large population sizes and more iterations, incurring heavy computational overhead.
2) **Extensive redundant performance evaluations**: During the current ENAS search process, each individual in the evolutionary population is evaluated. However, as the search progresses, some underperforming individuals have little to no impact on the global optimization in later iterations, but are still evaluated because they cannot be distinguished. These redundant evaluations not only fail to improve search quality but also lead to significant unnecessary computational consumption.

Therefore, focusing on the optimization of the search process and the reduction of redundancy evaluation, this paper proposes a Social Hierarchical-Guided Evolutionary Neural Architecture Search (SH-ENAS) algorithm, which aims to improve the global optimization ability of evolutionary search, reduce the population size and the number of iterations required in the search process, and thus reduce the consumption of computing resources.

The primary contributions of this paper are as follows:

1) Inspired by the social hierarchy, a novel population organization structure is designed and integrated with the evolutionary algorithm to effectively guide the subsequent search process.
    a) In this hierarchy, populations are divided into three subpopulations: upper, middle, and lower. By employing different crossover and mutation rate methods, each subpopulation is responsible for a different task: global optimization, potential exploration, and diversity maintenance, respectively.
    b) Based on the information from the upper subpopulations, a dynamic mutation operator selection and an operation type selection method are designed. This enables the search process to flexibly and adaptively adjust the size and structure of individuals under the guidance of high-quality individuals, thereby improving its search efficiency.
    c) Under the guidance of the number of iterations and the fitness change rate, a population reduction method is proposed for the lower subpopulation, which reduces unnecessary computational overhead by dynamically adjusting the population size.
2) To further reduce computational overhead, a progressive evaluation search method is proposed by introducing weight inheritance and validation-loss-guided early stopping operation to prevent unnecessary evaluations of architectures.

Experimental results on the CIFAR-10 and CIFAR-100 datasets demonstrate that SH-ENAS outperforms both classical manually designed networks and state-of-the-art NAS algorithms in terms

of accuracy and computational efficiency. Specifically, SH-ENAS achieves test errors of 2.50% on CIFAR-10 and 16.24% on CIFAR-100, surpassing all compared algorithms. Furthermore, SH-ENAS completes the search process with only 10 individuals and 12 iterations, reducing search costs to 0.69 and 0.83 GPU days, significantly outperforming most existing ENAS algorithms. These results underscore the comprehensive advantages of SH-ENAS in terms of accuracy, computational efficiency, and automation.

## 2    Related Work

In general, all kinds of existing NAS approaches face the challenge of high computational costs [17].For example, NASNet[20] employs an RL-trained controller RNN to iteratively optimize neural network cells, achieving near state-of-the-art (SOTA) performance on CIFAR-10. However, this method incurs extremely high computational costs; Han et al. [21] indicated that it requires over 1800 GPU days to search on ImageNet, making it impractical for resource-constrained environments. To mitigate computational costs, researchers have introduced surrogate models for rapid architecture evaluation, reducing evaluation overhead [22, 23]. Nevertheless, these approaches still rely on large-scale datasets with ground truth labels, limiting their practicality. Therefore, despite these improvements, RL-NAS is still computationally expensive and is far from being a practical solution for resource-constrained scenarios.Inspired by weight-sharing mechanisms, DARTS [18] proposed a gradient-based NAS method that constructs a supernet for efficient performance evaluation, significantly reducing search time. However, DARTS and its variants (e.g., PC-DARTS[24], Fair DARTS[25]) suffer from two major issues: (1) supernet construction requires domain expertise, making NAS less accessible to non-expert users and hindering true NAS automation; (2) performance collapse caused by parameter co-adaptation, where instability in subnet weights during supernet training deteriorates performance.

The earliest ENAS methods trace back to Genetic CNN, which encodes network architectures as fixed-length binary strings and requires 17 GPU days to train on CIFAR-10. Recent efforts have focused on optimizing performance evaluation to lower ENAS computational overhead. For instance, Qiu et al. [26] introduces performance predictor into ENAS, which employs a trained performance predictor to assess new architectures, reducing the need for full training and shortening search time to 1.9 GPU days. However, this method requires large-scale training data, making predictor training itself costly. Zhang et al.'s EvoNAS [27] allows offspring to inherit partial weights from parents, reducing evaluation time. However, its mutation operation is restricted to swapping units within individuals, limiting global search capabilities and preventing optimal architecture discovery. Real et al. proposed AmoebaNet [28], employing large-scale evolutionary search with population elimination methods to identify high-performance architectures, but the method still incurs substantial computational cost (3150 GPU days). Additionally, inspired by DARTS, Yang et al. [29] and Han et al. [21] incorporated supernet models into the ENAS process, enabling offspring networks to inherit supernet weights for accelerated evaluation. Nevertheless, supernet construction still demands domain knowledge and engineering efforts, increasing learning costs and contradicting NAS's goal of automation. Early stopping, proposed in [30, 31, 32], serves as an efficient mechanism to reduce computational overhead in evolutionary NAS by halting training when model performance plateaus. However, it may introduce evaluation bias due to varying convergence behaviors across architectures [33].

In summary, despite the progress made by NAS, it can be observed that how to significantly reduce the computational cost while pursuing high accuracy remains a huge challenge.

## 3    Method

### 3.1    Overall algorithm framework

Following the basic framework of ENAS as shown in Figure 1, SH-ENAS proposes innovative improvements in both the evolutionary search process and the performance evaluation process, indicated by shaded modules.

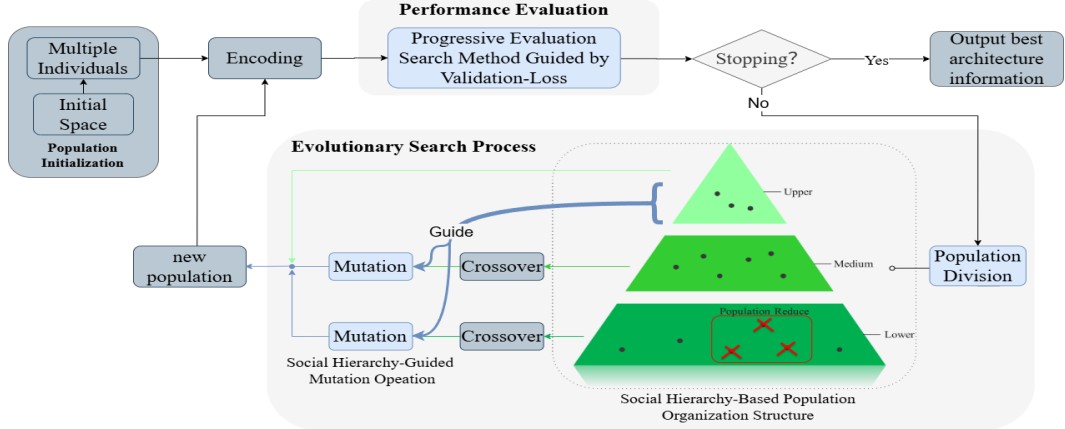

Figure 1: Overall algorithm framework

## 3.2 Social hierarchy-based population organization structure and its guided evolutionary methods.

### 3.2.1 Social hierarchy-based population organization structure

To enhance the global search ability of the GA algorithm and minimize the required population size, inspired by social hierarchy, we propose a new population organization structure and combine it with the GA algorithm to improve the evolutionary search efficiency of ENAS. In each iteration, we divided the whole population into three subpopulations - upper, middle, and lower - in descending order of fitness. The fitness is calculated using Equation 1, which considers both accuracy and model complexity. The higher the accuracy, the higher the fitness value of the individuals. And if the individual has a larger number of parameters, it will reduce its fitness. This encourages the development of lightweight, high-performance architectures.

$$Fitness(x) = \lambda \cdot \frac{acc(x) - acc_{min}}{acc_{max} - acc_{min}} + (1 - \lambda) \cdot (1 - \frac{parm(x) - parm_{min}}{parm_{max} - parm_{min}}) \qquad (1)$$

Where $\lambda$ is a coefficient balancing accuracy and parameter count, ranging from 0 to 1. $acc(x)$ represents the accuracy of the current architecture, with $acc_{min}$ and $acc_{max}$ denoting the minimum and maximum accuracy within the population, respectively. $parm(x)$ represents the parameter count of the current architecture, while $parm_{min}$ and $parm_{max}$ indicate the minimum and maximum parameter counts in the population.

The upper subpopulation contains the best architectures of the current population, which can provide positive guidance for the iterative update of the evolutionary algorithm, improving the efficiency of the search. The middle subpopulation includes structures with high potential. By designing and applying flexible and appropriate crossover and mutation operations, these structures can explore promising areas of the architectural space more efficiently.The lower subpopulation contains architectures that perform poorly within the population, and during the ENAS search process, the probability of crossover and mutation operations should be increased for these individuals to help them escape local optima, thus enhancing population diversity and search efficiency. Additionally, in later iterations of the evolutionary algorithm, computational overhead can be significantly reduced by stopping the evaluation of the worst individuals in this subpopulation.

In this social hierarchy-based structure, the members of each subpopulation are not fixed. After each iteration, they migrate between subpopulations based on updated fitness values, maintaining a stable number of individuals in each group.

The crossover rate and mutation rate of the evolutionary algorithm are vital factors in determining the degree of population change. In the current ENAS algorithm, all individuals in the population are basically given a consistent, fixed crossover rate and mutation rate, which allows all individuals to optimize in the same way. While in SH-ENAS, we design and assign different subpopulations with different crossover rates and mutation rates, as shown in Table 1, so that they can fulfill their respective

responsibilities, thereby improving the global search capability of the algorithm and reducing the needed population size.

Table 1: Adaptive crossover rates and mutation rates for subpopulations

| Subpopulation | Upper | Middle | Lower |
|---|---|---|---|
| Crossover Rate | $c_u = 0$ | $c_m = \lambda_m e^{-k_c \delta_t^2}(1 - \frac{t}{T})$ | $c_l = \lambda_l(1 - \frac{t}{T})$ |
| Mutation Rate | $m_u = 0$ | $m_m = \mu_m e^{-k_m \delta_t^2}(1 - \frac{t}{T})$ | $m_l = \mu_l(1 - \frac{t}{T})$ |

For the upper subpopulation, these individuals have high performance, and excessive modifications could degrade their performance. Thus, both crossover rate $c_u$ and mutation rate $m_u$ are set to zero, as shown in Table 1.

For the middle subpopulation, their fitness is at a suboptimal level, and better architectures can be obtained through crossover and mutation. Therefore, adaptive crossover rate $c_m$ and mutation rate $m_m$ are designed to more effectively explore and improve the population during evolution. As shown in Table 1, $\lambda_m$ is the crossover adjustment coefficient, and $k_c$ is a variance-based adjustment factor that regulates crossover intensity. $\delta_t^2$ represents the fitness variance within the middle subpopulation, $t$ is the current iteration, and $T$ is the maximum number of iterations. Similarly, $\mu_m$ is the mutation adjustment coefficient, ensuring controlled variation in the middle subpopulation, and $k_m$ is the corresponding variance-based adjustment factor for mutation.

For the lower subpopulation, adaptive crossover rate $c_l$ and mutation rate $m_l$ are designed. As shown in Table 1, higher adjustment coefficients $\lambda_l$ and $\mu_l$ encourage extensive recombination and mutation, enhancing diversity and escaping local optima. They gradually decrease over iterations to ensure stable and efficient convergence.

### 3.2.2 Social hierarchy-guided mutation

Mutation is a critical step in evolutionary algorithms, introducing diversity by randomly altering an individual's genetic structure to generate new architecture. Different from the majority of current ENAS, in SH-ENAS, we propose a dynamic mutation method (*Social hierarchy-guided mutation operator selection* and *Social hierarchy-guided Operation Type Selection*) to adaptively adjust the size and structure of the architecture under the guidance of the upper subgroup information to promote search efficiency and model adaptability.

In SH-ENAS, the architectures are encoded using the NASNet-style scheme [20]. Specifically, each cell is represented by node information and final connection information (contact). The node information comprises multiple operation-node pairs arranged sequentially, determining the network structure, with every two consecutive pairs corresponding to one output node. Furthermore, the supported operations include five major categories and nine subcategories (see Appendix C for details).

1) **Social hierarchy-guided mutation operator selection**
In current ENAS methods, the mutation operation can only be achieved by modifying the nodes, which greatly limits the search space that the algorithm can explore. In SH-ENAS, we supplement and design two additional mutation operations—addition and deletion—to expand the search space more comprehensively. Details of the mutation operations can be found in Appendix C.
In SH-ENAS, we propose an adaptive probability distribution for selecting mutation operators, which dynamically selects mutation operations according to the information derived from the upper subgroup, thereby improving the quality and efficiency of the search.
During population initialization, architectures are uniformly initialized with a node count $N_{init}$, calculated as:
$$N_{init} = \left\lfloor \frac{P}{N_u - N_l + 1} \right\rfloor \tag{2}$$
where $P$ represents the population size, $N_l$ and $N_u$ represent lower and upper bounds.
During the search process, the node distribution of the upper subpopulation is used to guide the mutation operation selection of the middle and lower subpopulations. Specifically, after each

iteration, the mode of the number of nodes in the upper subpopulation ($\bar{n}_{top}$), which reflects the scale of the current high-performance architecture, is calculated. Based on this, the probability functions for the three mutation operations (Add Operation: $p_a$, Modify Operation: $p_m$, Delete Operation: $p_d$) are defined as shown in Table 2.

Table 2: Mutation operation probabilities

| $\mathbf{n < \bar{n}_{top}}$ | | | $\mathbf{n = \bar{n}_{top}}$ | | | $\mathbf{n > \bar{n}_{top}}$ | | |
| --- | --- | --- | --- | --- | --- | --- | --- | --- |
| $p_a = \frac{\bar{n}_{top}-n}{\bar{n}_{top}-N_l}$, | $p_d = 0$, | $p_m = 1 - p_a$ | $p_a = 0$, | $p_d = 0$, | $p_m = 1$ | $p_a = 0$, | $p_d = \frac{n-\bar{n}_{top}}{N_u-\bar{n}_{top}}$, | $p_m = 1 - p_d$ |

When the architecture is small ($n < \bar{n}_{top}$), node addition is prioritized to enhance performance, with no deletion; when the architecture size is optimal ($n = \bar{n}_{top}$), only local modifications are allowed; and when the architecture is large ($n > \bar{n}_{top}$), node deletion is prioritized to reduce parameter overhead, again with no addition.

2) **Social hierarchy-guided Operation Type Selection**

Operation type is very important for ENAS, and the traditional ENAS algorithms suffer from inefficient search due to randomized operation type selection during mutation. This leads to slow convergence and suboptimal results. To address this issue, in SH-ENAS, by calculating the distribution probabilities of various operation types in the upper subpopulation, we propose a new Operation Type Selection method to guide the mutation of the middle and lower subpopulations towards the high-performance rapidly.

Specifically, we first compute the frequency of each operation type appearing in the upper subpopulation, denoted as: $T_o = [p_{o_1}, p_{o_2}, ..., p_{o_n}]$.

Where $p_{oi}$ represents the frequency of the $i$-th operation type, and $n$ is the total number of operation types. The frequency $p_{oi}$ is computed as:

$$p_{o_i} = \frac{o_i}{\sum_{i=0}^{n} o_i} \tag{3}$$

where $o_i$ denotes the number of occurrences of the $i$-th operation type in the upper subpopulation. Then, $T_o$ is used to guide operation type selection in the middle and lower subpopulations:

- **Add Operation**: The two most frequently occurring operation types in $T_o$ are selected to form a high-quality operation type set, denoted as $L_{good}$. When adding a new connection, a roulette wheel selection method is applied to determine the operation type from $L_{good}$, ensuring that high-quality operations are chosen more frequently, thereby improving search efficiency.
- **Modify Operation**: The two least frequently occurring operation types in $T_o$ are selected to form a low-quality operation type set, denoted as $L_{bad}$. When an operation from $L_{bad}$ exists in the current unit block, one connection containing such an operation is randomly removed, and a roulette wheel selection mechanism is then used to replace it with an operation from $L_{good}$.

### 3.3 Social hierarchy-guided population reduction

In the social hierarchy population, the lower subpopulation consists of a large number of individuals with relatively low performance. As the search proceeds, the performance of individuals in this subpopulation gradually stabilizes. However, these individuals continue to generate new candidates through crossover and mutation, whose evaluation leads to a substantial and increasingly unjustifiable rise in the overall search cost.

To address this issue, in SH-ENAS, we design a population reduction to improve algorithm efficiency. In the early stage, diversity is crucial, so no reduction is applied. As the population stabilizes, we gradually increase the reduction rate. The fitness variation rate serves as a key indicator of convergence speed. In the process of evolution, when the fitness changes relatively quickly, it indicates that the population is still very valuable overall, and it is not appropriate to reduce its individuals at this time. Conversely, if the fitness value changes slowly, indicating that the population has stabilized, reduction at this time will help to improve the efficiency of the search. The fitness variation rate $v$ is computed as:

$$v = \frac{f_t - f_{t-1}}{f_{t-1}} \, (t > 1) \tag{4}$$

where $f_t$ represents the average fitness at iteration $t$, and $f_{t-1}$ represents the average fitness at the previous iteration $t - 1$. Based on this, we define a probability function for population reduction:

$$P(t) = \begin{cases} 0, & t < \delta T \text{ or } v > v_{\text{threshold}} \\ \left(\frac{t}{T}\right)^{\alpha} \cdot \left(1 - \frac{v}{v_{\text{threshold}}}\right), & \text{otherwise} \end{cases} \tag{5}$$

Specifically, if the iteration count is below a predefined threshold $\delta T$ or if $v$ exceeds a threshold $v_{\text{threshold}}$, no reduction is performed. Here, $\delta$ is a reduction factor and $\alpha$ is a tuning factor. Otherwise, as iterations increase, the probability of reduction grows according to Equation 5, and the lower subpopulation size is dynamically adjusted based on $P(t)$, effectively reducing computational burden while maintaining search efficiency.

### 3.4 Progressive Evaluation Search Method Guided by Validation-Loss

In SH-ENAS, we propose a Progressive Evaluation Search method by introducing weight inheritance and validation-loss-guided early stopping operation. As shown in Figure 6, during the iterative process, a maximum training epoch limit $n$ is imposed, and a weight inheritance mechanism is utilized to retain the parameter information from unchanged parent models. Within each iteration, a dynamic early stopping criterion based on validation-loss is introduced: if the validation-loss remains below a predefined threshold $\delta$ for $m$ consecutive epochs, indicating that the model performance has stabilized, the current evaluation process is terminated to avoid unnecessary computational overhead. This method improves the utilization efficiency of computational resources while maintaining evaluation reliability.

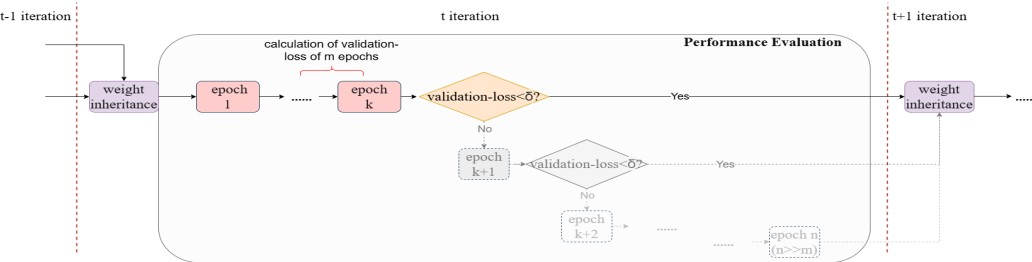

Figure 2: Illustration of the Progressive Evaluation Search method.

## 4 Experimental Results

This section evaluates the proposed SH-ENAS algorithm against classical manually designed networks and existing SOTA NAS algorithms on the CIFAR-10 and CIFAR-100 datasets. The analysis focuses on the comparison of accuracy, computational efficiency, as well as the comparison of population sizes and iteration counts with other evolutionary NAS methods. Ablation experiments are also conducted to verify the key improvement strategies of SH-ENAS.

All experiments were conducted using an Intel® Xeon Gold 5218R CPU, an NVIDIA Tesla V100 GPU, Ubuntu 18.04.2, Python 3.11.8, and PyTorch 2.2.2. Detailed experimental hyperparameters are provided in Appendix B.

### 4.1 Evaluation of SH-ENAS on CIFAR-10 and CIFAR-100

In this experiment, the compared algorithms include three prominent NAS categories: RL-NAS, Gradient-based NAS, and ENAS, representing current SOTA NAS approaches. The classification test error and GPU days of all compared algorithms are summarized in Table 3, where "–" indicates unpublished results, and all competitor results are sourced directly from their original papers.

As shown, SH-ENAS consistently outperforms manually designed architectures in test error. Specifically, on CIFAR-10, SH-ENAS achieves a lower error (2.50%) compared to DenseNet-BC (3.46%), MobileNet-V2 (4.26%), and ShuffleNet (9.13%). Similarly, on CIFAR-100, SH-ENAS attains

Table 3: Evaluation and comparison on CIFAR-10 and CIFAR-100.

| Architectures | Search Method | GPU Days | Params (M) | CIFAR-10 | CIFAR-100 |
|---|---|---|---|---|---|
| DenseNet-BC [34] | Manual Designed | - | 25.6 | 3.46 | 17.18 |
| MobileNet-V2 [35] | Manual Designed | - | 2.2 | 4.26 | 19.20 |
| ShuffleNet [36] | Manual Designed | - | 1.06 | 9.13 | 22.86 |
| NASNet-A [20] | RL | 1800 | 3.3 | 2.65 | 16.82 |
| Efficient NAS [23] | RL | 1 | 4.6 | 2.89 | 19.43 |
| MetaQNN [37] | RL | 100 | - | 6.92 | 27.14 |
| DARTS [18] | GD | 1 | 3.4 | 2.82 | 17.54 |
| DARTS-PT [38] | GD | 0.8 | 3.0 | 2.61 | - |
| Genetic CNN [39] | EA | 17 | - | 5.39 | 25.12 |
| CNN-GA [40] | EA | 35 | 2.9 | 3.22 | - |
| CNN-GA [40] | EA | 40 | 4.1 | - | 20.53 |
| AmoebaNet-A [28] | EA | 3150 | 3.2 | 3.34 | 18.93 |
| pEvoNAS-c10c [41] | EA | 1.41 | 3.0 | 2.73 | 17.77 |
| MOGIG-Net [42] | EA | 14 | 3.0 | 3.13 | 18.23 |
| CARS [29] | EA | 0.4 | 3.0 | 2.86 | 24.48 |
| ESENet-P [26] | EA | 1.4 | 4.26 | 3.90 | 24.97 |
| **SH-ENAS(*avg)** | **EA** | **0.69** 
 **0.83** | **3.4±0.12** 
 **4.32±0.45** | **2.57±0.07** 
 - | - 
 **16.46±0.22** |
| **SH-ENAS(*best)** | **EA** | **0.69** 
 **0.83** | **3.28** 
 **3.87** | **2.50** 
 - | - 
 **16.24** |

16.24%, surpassing DenseNet-BC (17.18%), MobileNet-V2 (19.20%), and ShuffleNet (22.86%). This highlights SH-ENAS's effectiveness as an automated alternative to manually designed networks.

Compared with SOTA NAS methods, SH-ENAS also achieves superior performance. On CIFAR-10, SH-ENAS outperforms NASNet-A (2.65%), Efficient NAS (2.89%), and DARTS (2.82%), while on CIFAR-100, it surpasses NASNet-A (16.82%), Efficient NAS (19.43%), and DARTS (17.54%). These results demonstrate the advanced performance of SH-ENAS, as well as its strong generalization and robustness.

SH-ENAS further excels in computational efficiency, requiring only 0.69 GPU days for CIFAR-10 and 0.83 GPU days for CIFAR-100, substantially lower than nearly all previously reported NAS methods. Although methods like CARS achieve lower GPU usage through supernet-based and expert-driven approaches, SH-ENAS maintains full automation, offering an optimal balance between computational efficiency and accuracy.

In summary, SH-ENAS demonstrates excellent accuracy, computational efficiency, and automation, positioning it as a promising and practical approach to neural architecture search.

## 4.2 Ablation experiment

To validate the effectiveness of the core components within our proposed SH-ENAS, we conducted a systematic ablation study. By selectively removing or replacing key methods, we analyzed their impact on algorithm performance. The experiments were conducted on the CIFAR-10 dataset with the same parameter settings. The experimental results are shown in Table 4.

Table 4: The ablation experiment of SH-ENAS

| Methods | | | | Results | | |
|---|---|---|---|---|---|---|
| Social Hierarchy-based Population Organization Structure | Social Hierarchy-guided Mutation | Population Reduction | Progressive Evaluation Search Method Guided by Validation-Loss | Accuracy | GPU Days | Params |
| × | × | × | × | 0.8693 | 12.50 | 1.82 |
| ✓ | ✓ | ✓ | ✓ | 0.9047 | 0.69 | 1.27 |
| × | ✓ | ✓ | ✓ | 0.8900 | 0.76 | 1.89 |
| ✓ | × | ✓ | ✓ | 0.8930 | 0.69 | 1.56 |
| ✓ | ✓ | × | ✓ | 0.9038 | 0.86 | 1.32 |
| ✓ | ✓ | ✓ | × | 0.9068 | 8.10 | 1.29 |

The results indicate that enabling all proposed methods (row 2) achieves the highest accuracy (90.47%), lowest computational cost (0.69 GPU days), and smallest parameter count (1.27M).

Ablation of individual methods leads to decreased accuracy and/or increased computational cost, highlighting their importance.

**Social Hierarchy-based Population Organization Structure**: Removing this method (row 3) reduces accuracy to 89.00% and increases GPU days to 0.76, underscoring its role in improving generalization and search efficiency.

**Social Hierarchy-Guided Mutation**: Disabling this method (row 4) reduces accuracy to 89.30% and increases parameters to 1.56M, indicating its effectiveness in discovering compact and efficient architectures.

**Population Reduction**: Omitting this approach (row 5) slightly lowers accuracy (90.38%) while significantly increasing GPU days (0.86), illustrating its contribution to computational efficiency.

**Progressive Evaluation Search Method Guided by Validation-Loss**: Removing this method (row 6) substantially increases GPU days (8.10) despite slightly improved accuracy (90.68%), confirming its critical role in reducing computational cost.

**Baseline Configuration**: Disabling all proposed methods (row 1) leads to the lowest accuracy (86.93%), highest computational cost (12.50 GPU days), and largest parameter count (1.82M), clearly demonstrating the combined value of the proposed methods.

Overall, this ablation study validates that integrating these methods achieves an optimal balance among accuracy, computational efficiency, and model complexity, confirming SH-ENAS as an effective neural architecture search approach.

## 4.3 Comparison of SH-ENAS with Other ENAS Algorithms in Terms of Population Size and Iteration Count

To evaluate SH-ENAS's efficiency, we compare it with Genetic CNN, CNN-GA, pEvoNAS-c10c, and CARS in terms of population size and iteration count, as shown in Figure 3.

As shown in the figure, SH-ENAS resides in the bottom-left region, requiring fewer individuals and iterations. It uses only 10 individuals—half the size of typical methods—and converges within 12 iterations, compared to over 20 for others.

This highlights SH-ENAS's ability to maintain optimization quality while reducing resource demands, confirming its efficiency and well-balanced design.

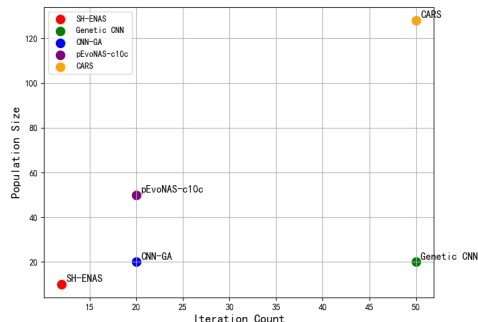

Figure 3: Comparison of population size and iteration count.

## 5 Conclusion

**Conclusion**: This study introduces Social Hierarchy-Guided Evolutionary Neural Architecture Search (SH-ENAS), leveraging social dynamics to enhance exploration and exploitation during architecture search. Experiments demonstrate SH-ENAS outperforms existing NAS methods in accuracy, efficiency, and robustness.

**Impact**: SH-ENAS contributes theoretically and practically by integrating structured social dynamics into evolutionary NAS. It can extend to other domains like reinforcement learning and NLP, supporting resource-efficient NAS, especially in limited-resource scenarios, potentially influencing future algorithmic designs.

**Limitation**: Although effective, SH-ENAS primarily targets CNN-based image classification tasks. Future research should explore broader applications and scalability via distributed computing or hardware-aware NAS, addressing real-world constraints like hardware limitations and energy efficiency.

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

# A Weight inheritance

This subsection presents an example of weight inheritance, including crossover and mutation operations.

## A.1 Crossover weight inheritance

When generating the structure of an offspring individual through crossover, the training parameters of the parent networks are directly used as the initial weights for the corresponding nodes in the offspring. Since the crossover operation does not introduce any new nodes, all node parameters of the offspring can be directly inherited from the parents without the need for random reinitialization of weights.

Figure 4 illustrates an example of offspring generation through weight inheritance in a crossover operation. In the cell-based encoding scheme, each edge represents an operation, meaning that each edge corresponds to a specific weight. Suppose individuals $p_1$ and $p_2$ are selected as parents for crossover. A crossover is performed between the fourth and fifth genes to produce two offspring, denoted as $c_1$ and $c_2$. Let $\theta$ represent the set of inheritable parameters received by the offspring. It is evident that all weights in the offspring are inherited from the parents.

As shown in Figure 4, offspring $c_1$ inherits four operations (highlighted in white boxes) from parent $p_1$, while the remaining four operations (yellow boxes) are inherited from parent $p_2$. Accordingly, weights $w_1, w_2, w_3, w_4$ are assigned from $p_1$, and $w_{13}, w_{14}, w_{15}, w_{16}$ are assigned from $p_2$ to the corresponding edges in $c_1$. Similarly, offspring $c_2$ inherits four operations (white boxes) from parent $p_2$, and the remaining four operations (yellow boxes) from parent $c_1$. The corresponding weights $w_9, w_{10}, w_{11}, w_{12}$ and $w_5, w_6, w_7, w_8$ are assigned from $p_2$ and $p_1$, respectively, to the appropriate edges in $c_2$.

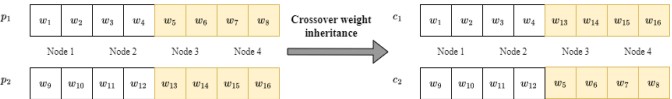

Figure 4: The example of crossover weight inheritance

## A.2 Mutation weight inheritance

When generating the structure of an offspring individual through mutation, the original architecture is altered, preventing the offspring from fully inheriting all weights from its parent. However, the offspring can inherit the weights of components that remain unchanged during the mutation process.

Figure 5 illustrates an example of offspring generation via weight inheritance under mutation. Assume that the parent individual $p$ undergoes three types of mutation operations—addition, modification, and deletion—resulting in three offspring individuals, $c_1$, $c_2$, and $c_3$, respectively. Let $w_*$ denote the set of weights inherited by the nodes. It can be observed that the offspring individuals inherit the weights associated with the unchanged parts of the parent architecture.

As shown in Figure 5(a), during the addition operation, offspring $c_1$ inherits eight operations from $p$ (highlighted in yellow boxes), while two new operations (highlighted in white boxes) are introduced through the added node. Consequently, weights $w_1$ to $w_8$ are transferred from $p$ to $c_1$, while the new weights $w_9$ and $w_{10}$ are randomly initialized.

Figure 5(b) illustrates the case of the modification operation. Offspring $c_2$ inherits seven operations from $p$ (yellow boxes), while one operation (white box) is altered. In this scenario, weights $w_1$ to $w_6$ and $w_8$ are inherited from $p$, whereas the original weight $w_7$ corresponding to the modified operation is replaced with a newly initialized weight $w_7'$.

Figure 5(c) demonstrates the deletion operation. Offspring $c_3$ inherits six operations from $p$ (yellow boxes), with the remaining components removed. Accordingly, weights $w_1$ to $w_6$ are directly inherited from the parent $p$.

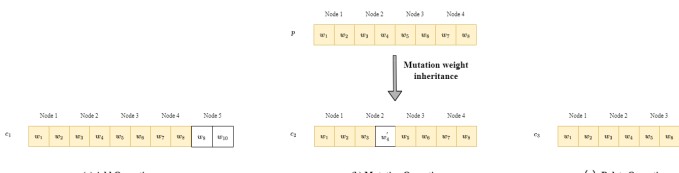

Figure 5: The example of mutation weight inheritance

# B  Parameters Determination

SH-ENAS involves several key parameters, such as the crossover adjustment factor of the middle subpopulation. To determine these parameters, this section conducts experiments based on the CIFAR-10 dataset. In the experiment, the population size is set to 10 individuals, each with an architecture comprising an 8-layer neural network (consisting of 6 normal cells and 2 reduction cells). The algorithm runs for 12 iterations. The focus of the experiment is to identify the core parameters of SH-ENAS, including the subpopulation division ratio, and the adaptive crossover and mutation rate adjustment coefficients.

## B.1  Subpopulation division ratio

In SH-ENAS, the population is divided into three subpopulations, each representing a different region of the network architecture search space. The division of subpopulations significantly affects both the efficiency of the search process and the performance of the final network architecture. To determine the optimal subpopulation division ratio, four sets of experiments were designed to compare the search results under different ratios. The experimental results are as follows: As shown in the table,

Table 5: Subpopulation Division Experiment Results

| Upper: Middle: Lower | Acc | GPU Days | Params/M | FLOPS/B |
|---|---|---|---|---|
| 1:4:5 | 0.8976 | 0.68 | 1.04 | 0.17 |
| 1:2:2 | 0.9047 | 0.69 | 1.27 | 0.21 |
| 1:1:3 | 0.8924 | 0.68 | 1.66 | 0.29 |
| 3:3:4 | 0.8965 | 0.72 | 1.53 | 0.25 |

when the subpopulation division ratio is 1:2:2, the algorithm performs the best in terms of accuracy (Acc), achieving 0.9047. Furthermore, the GPU days consumed at 0.69 is quite reasonable compared to other division ratios. Under this ratio, the network architecture's parameter count and FLOPS are relatively low, with values of 1.27 Params/M and 0.21 FLOPS/B, indicating a good balance between performance and computational efficiency. In contrast, the 1:4:5 and 1:1:3 ratios, although requiring fewer GPU days than the 1:2:2 configuration, show significant differences in accuracy. The 3:3:4 ratio configuration performs the worst, with both lower accuracy and higher computational resource consumption.

In conclusion, the 1:2:2 subpopulation division ratio provides the best balance between search efficiency and the final network architecture's performance. This ratio effectively utilizes computational resources, achieves higher accuracy, and maintains the network architecture's parameter count and computational complexity within a reasonable range. Therefore, the 1:2:2 subpopulation division ratio is selected as the optimal configuration for this algorithm.

## B.2  Adaptive Crossover and Mutation Rate Adjustment Coefficients

In SH-ENAS, the adaptive crossover and mutation rate adjustment coefficients significantly influence the extent of updates within the population. These adjustment coefficients include four key parameters: the crossover rate adjustment coefficient for the middle subpopulation, the mutation rate adjustment coefficient for the middle subpopulation, the crossover rate adjustment coefficient for the lower subpopulation, and the mutation rate adjustment coefficient for the lower subpopulation. These coefficients determine the intensity of crossover and mutation operations within the middle and lower

subpopulations, which in turn impacts the search efficiency and final performance of the algorithm. To determine the optimal configuration of these parameters, experiments were conducted using various combinations of crossover and mutation rate adjustment coefficients, and their impact on performance was evaluated. The experimental results are shown in the following table: From the

Table 6: Adaptive Crossover and Mutation Rate Adjustment Coefficients Experiment Results

| $(\lambda_m, \lambda_l, \mu_m, \mu_l)$ | Acc | GPU Days | Params/M | FLOPS/B |
|---|---|---|---|---|
| (0.65, 0.75, 0.7, 0.8) | 0.8976 | 0.74 | 1.04 | 0.17 |
| (0.45, 0.55, 0.5, 0.6) | 0.9047 | 0.69 | 1.27 | 0.21 |
| (0.25, 0.35, 0.3, 0.4) | 0.9043 | 0.68 | 2.38 | 0.39 |

experimental results in Table 6, it is evident that when the crossover and mutation rate adjustment coefficients are configured as (0.45, 0.55, 0.5, 0.6), the algorithm achieves the highest accuracy of 0.9047, with a GPU days consumption of 0.69. Although the parameter count and FLOPS of the resulting architecture are relatively high, this configuration offers a good balance between accuracy and computational efficiency.

In contrast, when the coefficients are set to (0.25, 0.35, 0.3, 0.4), the GPU days consumed are lower, but the resulting architecture has the largest parameter count and FLOPS, and the accuracy is slightly lower than the previous configuration. This indicates that lower crossover and mutation intensities limit the exploration of the search space.

When the coefficients are set to (0.7, 0.8, 0.65, 0.75), the crossover and mutation operations are more intense, resulting in the lowest accuracy and the highest GPU days consumed. This suggests that excessively high crossover and mutation intensities can lead to instability in the search process, negatively impacting final performance.

In conclusion, the (0.45, 0.55, 0.5, 0.6) configuration of crossover and mutation rate adjustment coefficients provides the highest accuracy while consuming a reasonable amount of GPU days. Therefore, this configuration is chosen for algorithm optimization.

## B.3 Other Parameters in SH-ENAS

In addition to the key parameters discussed above, SH-ENAS also includes other parameters that influence the search efficiency and overall performance of the algorithm. The specific values for these parameters used in SH-ENAS are detailed in Table 7.

Table 7: Other Parameters in SH-ENAS

| Parameter | Value |
|---|---|
| Population Size | 10 |
| Iteration Count | 12 |
| Fitness Function Weight Coefficient ($\lambda$) | 0.95 |
| Epoch Limit per Iteration in Early Stopping | 15 |
| Patience in Early Stopping | 5 |
| Early Stopping Threshold ($\delta$) | 0.001 |
| Speed Threshold for Population Reduction ($v_{\text{threshold}}$) | 0.06 |
| Adjustment Factor in Population Reduction ($\alpha$) | 0.8 |
| Learning Rate for Architecture Search | 0.0027 |
| Optimizer for Architecture Search | SGD |
| Training Batch Size during Architecture Search | 96 |
| Validation Batch Size during Architecture Search | 96 |

These parameters include population size, maximum iteration count, and fitness function adjustment coefficient, among others. Although these parameters impact algorithm performance, their optimal configurations can vary depending on specific tasks and have broader implications. Therefore, instead of conducting individual tuning experiments for these parameters, their values were determined based on empirical settings from related literature.

## C  Mutation Operations in SH-ENAS

Existing ENAS methods typically employ NASNet-style encoding, where each cell is defined by node pairs comprising operation and input-node information, significantly limiting the mutation space due to a fixed number of nodes. In contrast, SH-ENAS introduces a variable-length encoding scheme through three distinct mutation operations: addition, modification, and deletion, as demonstrated by the following examples.

Given an initial encoded cell:

$$[(0, 0), (4, 1), (8, 1), (8, 2), (2, 0), (4, 2), (3, 2), (4, 3)],$$

the three mutation operations are defined as:

- **Add Operation:** A new node is appended at the end of the cell. For example, adding node pairs (5, 5) and (9, 4).
- **Modify Operation:** An existing operation type within a node pair is changed. For instance, changing the 5th node pair operation from a skip connection (2) to a dilated 5×5 convolution (6).
- **Delete Operation:** The last node pairs are removed. For example, deleting the node pairs (3, 2) and (4, 3).

A graphical illustration of these mutation operations is provided in Figure 6.

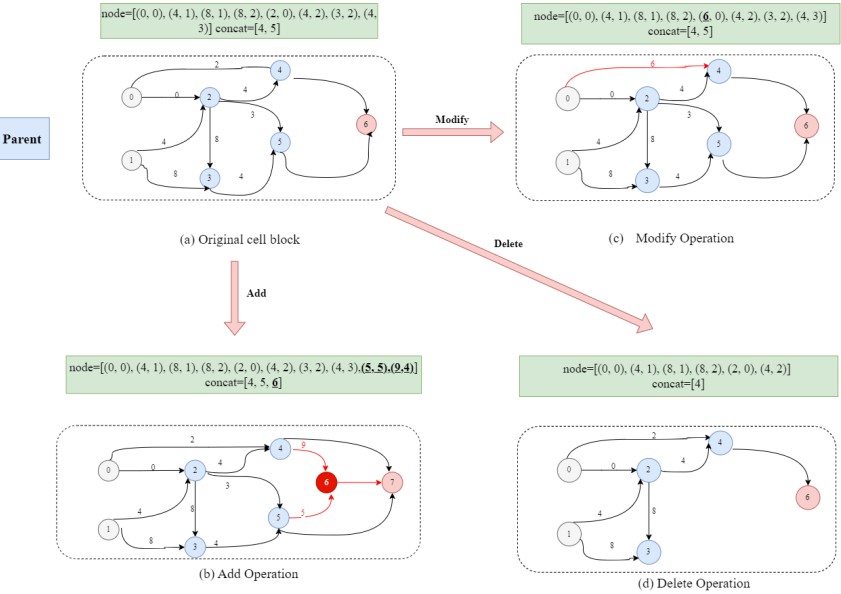

Figure 6: Examples of the three mutation operations in SH-ENAS

## D  Operation Types in SH-ENAS

Mutation is essential in evolutionary algorithms to introduce diversity by randomly altering architectures. SH-ENAS implements a dynamic mutation strategy guided by information from the upper subpopulation, promoting efficiency and adaptability. The specific operation types used in SH-ENAS are detailed in Table 8, categorized into five main groups with nine distinct operations.

Table 8: Mutation operation types in SH-ENAS

| Operation Type | Specific Operation | Encoding |
|---|---|---|
| Basic Pooling Operation [43] | Average Pooling | 0 |
|  | Max Pooling | 1 |
| Skip Connection [44] | – | 2 |
| Depthwise Separable Convolution Block [45] | Depth-wise separable 3×3 | 3 |
|  | Depth-wise separable 5×5 | 4 |
|  | Depth-wise separable 7×7 | 5 |
| Dilated Convolution Block [46] | Dilated 5×5 | 6 |
|  | Dilated 7×7 | 7 |
| Inverted Residual Block [35] | Inverted residuals 3×3 | 8 |
|  | Inverted residuals 5×5 | 9 |

