# OpenReview forum: "Social Hierarchy-Guided Evolutionary Neural Architecture Search for Efficient and Automated Design"
_NeurIPS.cc/2025/Conference — Submitted to NeurIPS 2025_

### Official Review · Reviewer_LfUB · 2025-06-13

**Clarity:** 3
**Significance:** 3
**Originality:** 3
**Rating:** 3
**Confidence:** 4

**Summary:**

The paper presents SH-ENAS, a hierarchical evolutionary NAS that finds competitive CIFAR-10/100 architectures in under one GPU-day by assigning different crossover/mutation settings to upper, middle, and lower tiers and by using lightweight weight-inheritance evaluation.

**Questions:**

1. Have you evaluated SH‑ENAS on ImageNet‑16‑120 or full ImageNet? If not, what barriers prevent scaling?

2. Could you report mean ± std over ≥ 3 independent searches to show robustness?

**Ethical Concerns:**

["NO or VERY MINOR ethics concerns only"]

**Limitations:**

Yes

**Quality:**

2

**Strengths And Weaknesses:**

Strengths:

1.	The paper introduces a hierarchical population schedule that dynamically varies crossover and mutation rates to balance exploration and exploitation.

2.	It matches strong NAS baselines on CIFAR-10/100 while using only about 0.7 GPU-day of compute.

3.	Ablation experiments cleanly show the individual impact of hierarchy, adaptive operator selection, and population-reduction.

Weaknesses:

1.	Both hierarchical population control and weight-inheritance evaluation are known ideas in evolutionary NAS; the contribution is primarily an incremental combination rather than a fundamentally new algorithmic insight.

2.	All experiments are confined to CIFAR-10/100; without ImageNet-scale or non-vision benchmarks, it is unclear whether the same efficiency and accuracy gains persist on larger or different tasks.

3.	Results are reported from single-seed searches only; the absence of multiple runs and standard-deviation statistics leaves open the question of robustness and statistical significance.

4.	Although the text claims the code will be open-sourced, the submission does not include a repository link or supplementary zip, preventing reviewers from verifying implementation details immediately.

5.	The search space is limited to classic CNN cell structures; the paper gives no evidence that the hierarchy strategy transfers to modern backbones such as Vision Transformers or to architectures for detection and segmentation.

---

### Official Review · Reviewer_EhfG · 2025-06-15

**Clarity:** 2
**Significance:** 2
**Originality:** 2
**Rating:** 2
**Confidence:** 4

**Summary:**

This paper introduces SH-ENAS, an evolutionary NAS algorithm inspired by social hierarchies. It employs a novel population structure divided into three sub-populations, upper, middle, and lower, to balance global optimisation, exploration, and diversity maintenance. Additionally, SH-ENAS incorporates a progressive evaluation strategy featuring weight inheritance and validation-loss-based early, significantly reducing computational overhead. Empirical results on CIFAR-10/-100 demonstrate SH-ENAS achieving state-of-the-art accuracy while substantially reducing computational resources compared to exisiting methods.

**Questions:**

1. Can authors test SH-ENAS on more challenging datasets beyond CIFAR-10/-100, such as ImageNet?
2. Can authors provide theoretical analysis or intuition as to why the social hierarchy-based structure specifically improves search efficiency?
3. How sensitive is SH-NAS performance to variations in population size and the number of iterations?

**Ethical Concerns:**

["NO or VERY MINOR ethics concerns only"]

**Limitations:**

The paper has some limitations but insufficiently addresses, such as:
1) lack of evaluations on more challenging tasks, nor comparison with more recent work.
2) the adaptive parameters, e.g., mutation and crossover rates, could lead to significant reproducibility challenges. Authors should provide clearer guidelines on tuning these adaptive mechanisms or demonstrating the parameter robustness of the proposed method.

**Quality:**

2

**Strengths And Weaknesses:**

**Strengths**
1. The proposed social hierarchy-based population management strategy is interesting and creative within evolutionary NAS field, providing structured guidance for evolution processes.
2. The paper is well structured and easy to follow.

**Weaknesses**
1. The paper primarily focuses on empirical observations without rigorous theoretical explanations about why the social hierarchy approach specifically enhances search performance or efficiency.
2. The evaluations primarily conducted on the CIFAR-10/-100 dataset, which are relatively simple. Applicability to more challenging tasks like ImageNet, or different domains, is not explored, this limiting the claims of generalisation.
3. Most of the compared methods in Table 3 are a bit outdated, more recent and advanced evolutionary NAS methods such as QE-NAS [1], ZiCo [2], or SWAP [3] etc., are not included in the comparison, nor discussed in the related work.


*[1] Z. Sun, C. Ge, J. Wang, M. Lin, H. Chen, H. Li, and X. Sun, “Entropy-driven mixed-precision quantization for deep network design.”, NeurIPS 2022.*

*[2] G. Li, Y . Y ang, K. Bhardwaj, and R. Marculescu, “Zico: Zero-shot NAS via inverse coefficient of variation on gradients.”,  ICLR 2023.*

*[3] Y. Peng, A Song, H. Fayek, V. Ciesielski, and X. Chang. "SWAP-NAS: Sample-wise activation patterns for ultra-fast NAS.", ICLR 2024.*

---

### Official Review · Reviewer_cWbd · 2025-07-01

**Clarity:** 2
**Significance:** 3
**Originality:** 3
**Rating:** 3
**Confidence:** 4

**Summary:**

This paper proposes an evolutionary architecture search method based on social hierarchy population organization, where the population is split into upper, middle and lower subpopulations. This social hierarchy structure guides both mutation operation selection and operation type decisions during the evolutionary search process. To improve efficiency, the method incorporates a population reduction strategy that avoids unnecessary evaluations of low-performing individuals. Additionally, an early stopping criterion based on validation loss is employed, that significantly accelerates the search with minimal impact on final performance. Experiments on CIFAR-10 and CIFAR-100 demonstrate the effectiveness of the proposed approach compared to several manually designed models as well as architectures discovered by various SOTA NAS methods.

**Questions:**

- The CIFAR-10 results reported in Tables 3 and 4 seem inconsistent. Could the authors clarify the source of this discrepancy and indicate whether different search spaces were used in these experiments?
- How were the hyperparameters used in the method (e.g. those in equations 1,2,5, and Table 1) selected in practice, and how sensitive are the results to these choices?
- It appears that all methods in Table 3, except SH-ENAS, transfer architectures from CIFAR-10 to CIFAR-100. Have the authors also tried to transfer their CIFAR-10 architecture to CIFAR-100 in the same way?

**Ethical Concerns:**

["NO or VERY MINOR ethics concerns only"]

**Limitations:**

yes

**Paper Formatting Concerns:**

I did not notice any major formatting issues.

**Quality:**

2

**Strengths And Weaknesses:**

Strengths:

- The paper presents a novel evolutionary NAS idea based on a socially hierarchical population structure. This approach opens a promising direction for bringing evolutionary NAS methods closer to gradient-based methods in terms of search efficiency, while maintaining strong performance.
- The ablation studies are well-executed. They effectively isolate the contribution of each component in the proposed method, providing empirical support for the overall design of the method.

Weaknesses:

- Experiments are limited to CIFAR-10 and CIFAR-100, which makes it difficult to assess the method’s generalizability and effectiveness on more diverse tasks.
- The search space appears to be closely related to DARTS-like cell-based search spaces, which the paper acknowledges as requiring domain expertise. This raises concerns about the method’s applicability to richer and more flexible search spaces. Moreover, the full architecture, including the backbone and the placement and number of cells, is not clearly described in the main paper. In particular, Appendix D, which details the operation types, should be referenced in the main text.

---

### Official Review · Reviewer_BcQw · 2025-07-03

**Clarity:** 2
**Significance:** 2
**Originality:** 2
**Rating:** 2
**Confidence:** 4

**Summary:**

This paper proposes SH-ENAS, a novel evolutionary neural architecture search (ENAS) algorithm guided by a social hierarchy-inspired population structure. SH-ENAS introduces effective evolutionary operations and a progressive evaluation strategy that incorporates weight inheritance and early stopping to reduce computational costs. Experiments on CIFAR-10 and CIFAR-100 show that SH-ENAS achieves competitive accuracy (2.50% and 16.24% test error, respectively) while requiring significantly fewer resources, only 10 individuals, 12 iterations, and under 1 GPU day, which highlighting its effectiveness in both accuracy and efficiency.

**Questions:**

as my concerns

**Ethical Concerns:**

["NO or VERY MINOR ethics concerns only"]

**Limitations:**

yes

**Quality:**

2

**Strengths And Weaknesses:**

Strengths:
1. The proposed SH-ENAS introduces a novel coarse-to-fine evolutionary algorithm for neural architecture search (NAS). It guides the search process by dividing the search space into three hierarchical populations: upper, middle, and lower levels.
2. SH-ENAS adaptively determines the values of certain hyperparameters, allowing the method to dynamically adjust architecture size and structure. This contributes to an expanded search space and improved efficiency.

Major Concerns:
1. The experimental evaluation is insufficient. The main experiments are limited to searches on CIFAR-10 and CIFAR-100, which is inadequate within the NAS community. NAS methods are often heavily influenced by randomness due to stochastic sampling and other factors. As a result, merely reporting the performance of a single best-found architecture on specific datasets such as CIFAR lacks persuasive power.
2. Given that the method adopts a graph-based encoding similar to NASNet, it is unclear why experiments were not conducted on additional cell-based search spaces.
3. Following the above points, the experiments lack standard benchmarks such as NASBench-101, NASBench-201, or NASBench-301. The evaluation should also include metrics like Kendall’s Tau correlation or Spearman correlation to support the claim that the proposed search algorithm possesses a global understanding capability.
4. The algorithm appears to involve numerous critical hyperparameters. Currently, only partial results are provided in Tables 6 and 7. However, there is a lack of comprehensive ablation studies. Each hyperparameter's influence on the proposed evolutionary algorithm should be thoroughly analyzed rather than simply comparing settings with and without certain components.

Minor Concerns:
1. Figures 1 and 2 appear distorted, and the font in Figure 2 is too small and should be enlarged.
2. For Figures 1, 2, and 3, please consider using vector graphics formats such as SVG or PDF. Raster images such as PNG or JPG result in reduced clarity.

Comment:
The proposed clustering-based approach is interesting, but the design is overly complex and involves a large number of hyperparameters. I recommend that the authors first attempt to simplify the method. If not, then the ablation study must be detailed and comprehensive. More importantly, the current experimental setup is lacking in key aspects. For a newly proposed NAS method, this is a critical issue. It is essential to conduct experiments on standard NAS benchmarks such as NASBench-101, NASBench-201, NASBench-301, or others to enhance credibility. If the above-mentioned key experiments are supplemented, this work could be highly promising. Based on the current status, I am inclined to recommend rejection.

---

### Decision · Program_Chairs · 2025-09-17

**Decision:**

Reject

**Comment:**

This paper presents SH-ENAS, a novel and efficient evolutionary NAS algorithm. While the core "Social Hierarchy" concept is interesting, the submission is critically undermined by insufficient experimental validation, as all reviewers noted. The evaluation is confined to CIFAR-10/100, omitting standard benchmarks (e.g., NASBench-101/201) and larger datasets like ImageNet. Furthermore, it lacks comparisons to recent methods and robust, multi-run results. Notably, the authors did not provide a rebuttal to address these significant concerns. The recommendation is therefore **Reject**. The authors are advised to significantly expand their experiments for a future submission.